# The Impact of Physical Activity on Memory Loss and Concentration in Adults Aged 18 or Older in the U.S. in 2020

**DOI:** 10.3390/ijerph21091193

**Published:** 2024-09-09

**Authors:** Serena C. L. Buchwald, Daniel Gitelman, Dins Smits, Pura E. Rodriguez de la Vega, Noël C. Barengo

**Affiliations:** 1Department of Medical Education, Herbert Wertheim College of Medicine, Florida International University, Miami, FL 33199, USA; serena.buchwald@gmail.com (S.C.L.B.); dgite001@fiu.edu (D.G.); rodrigup@fiu.edu (P.E.R.d.l.V.); 2Faculty of Medicine, Riga Stradins University, LV-1007 Riga, Latvia; dins.smits@gmail.com; 3School of Medicine, National University of Mar del Plata, Mar del Plata 7600, Argentina

**Keywords:** dementia, memory loss, concentration, physical activity, adults

## Abstract

This cross-sectional study used secondary data from the USA 2020 National Health Interview Survey database. The goal of this study is to outline the impact physical activity has on cognition and mental ability. The reason we chose to pursue this research was a result of the exponentially growing weight of economic and emotional burden caused by cognitive impairments and diseases. The main outcome was whether individuals experienced dementia symptoms such as memory loss and difficulty concentrating. The main exposure was following physical activity guidelines (none, strength only, aerobic only, both). The confounders included age, sex, region, heart disease status, smoking status, drinking status, and depression status. The sample is composed of 30,119 USA adults aged 18 or older. Of those participants, 46% were male and 54% were female. By age, 96% were 18–84 years old, and approximately 4% were 85 and older. Logistic regression analysis was used to calculate odds ratios (ORs) and 95% confidence intervals (CIs). There was a statistically significant association between difficulty following physical activity guidelines and cognitive difficulties. Those who met aerobic only increased the odds of cognitive difficulty by 52% (OR 1.52; 95% CI: 1.34–1.74) compared with those who met both criteria. Those who met the strength criteria had 1.7 greater odds of cognitive difficulties (OR 1.70; 95% CI: 1.42–2.02) than those who met both criteria. Those who met neither of these guidelines had almost threefold greater odds of having cognitive difficulties (OR 2.64; 95% CI: 2.36–2.96) than those who met both guidelines. Researchers and healthcare providers should collaborate to encourage meeting these guidelines and addressing barriers preventing people from being physically active, such as physical limitations and access to safe recreational spaces. Future studies should address the health disparities regarding physical activity.

## 1. Introduction

Dementia is characterized by a decline in cognition and involves memory loss and difficulty concentrating, according to Kumar et al. [1]. This decline in cognition is severe enough to disrupt one’s normal daily functioning and quality of life [2]. The exact causes of dementia vary depending on genetics and environmental and lifestyle factors. The prevalence of forms of dementia is increasing. More than 7 million adults aged 65+ were living with dementia in 2020 [3]. Dementia and cognitive difficulties negatively impact quality of life. It takes hold of physical, psychological, social, and economic aspects of people’s lives [4]. According to the World Health Organization (WHO), dementia is the seventh leading cause of death globally. It simultaneously acts as one of the major causes of disability and dependency on the global scale as well. Furthermore, the economic burden of dementia is staggering. The global economic burden totaled approximately USD 1.3 trillion. These costs were primarily attributed to informal care providers. Conducting greater research on this subject would yield significant advancements in the level of care and efficiency supplied globally. It would perpetuate advancements in care quality, accessibility, cost-efficiency, and overall life improvements for those impacted by dementia and the symptoms accompanied by it. Further expanding upon these conditions would resolve the stigmatization developed by those underinformed and tear down social barriers that limit the ease of access to those impacted. Cognitive difficulties are sensitive and complex, and elaborating upon these difficulties can supply much-needed education, healthcare services, access to treatment, and continued growth in social understanding of the conditions.

Physiologically, dementia can present as cortical atrophy, a decrease in brain volume, and the presence of neurofibrillary tangles and plaques. The degree of clinical illness and cognitive decline is inversely correlated to the number of tangles [5]. As a result of the neuronal cell loss in dementia, there is no current treatment. However, there are ways to manage and prevent the early onset of symptoms, with the main goal of this management being the delay of cognitive decline and reduction in cognitive suffering [5]. With dementia and cognitive difficulties appearing more regularly in older adults, the importance of physical activity cannot be understated. One study specifically found that being physically active can improve lifestyle factors such as basic cognition, independent functioning, and overall psychological health [6]. Physical activity was proven to especially benefit executive functioning and overall memory retention in adults. This applies more sensitively to the older population, as it is more difficult for older adults to execute proper physical exercise; however, the benefits of such practices have been proven to undeniably improve their quality of life, cognitive ability, and health as a whole.

A study by Castaño showed that physical activity combined with cognitive training increased brain-derived neurotrophic factor levels. Moreover, an increase in brain-derived neurotrophic factor (BDNF) was positively associated with improved cognition [7]. An increase in brain biomarkers caused by physical activity was also reported to be associated with improved cognition [8] and improved brain function [9]. A systematic review by Chang et al. on the effect of resistance exercise training on cognitive function in healthy older adults revealed that intervention designs involving the use of one repetition maximum load of 60–80% with approximately seven movements in two sets separated by 2 min of rest at least twice per week for 2–12 months (usually 6 months) may positively affect cognition, including information-processing speed, attention, memory formation, and specific types of executive function [8].

However, one major limitation of previous studies is that most of them did not include a representative sample of the general population [10,11]. Additionally, these previous studies assessed changes in healthy lifestyle choices in middle-aged adults (50–60 years of age), and data on the association between physical activity and cognitive function in older adults are scarce. The WHO recommends that older adults should engage in 150–300 min of moderate-intensity aerobic physical activity per week combined with strength exercise twice per week. However, there is little scientific evidence available on whether meeting these guidelines is associated with memory loss or difficulties concentrating in older adults. Beyond those limitations, many other examples plague the research surrounding dementia and stunted cognition currently. The primary issue is the lack of interconnected literature in the domains that directly benefit from physical activity. Many pieces of literature regarding our topic generalize the benefit of physical activity with overall health improvement in many facets but fail to supply feasible interventions for those impacted. Today, this problem is more focused on solutions to heart health without connecting the benefits to cognitive health as well. If more research was committed to developing interventions that both older and younger adults could benefit from to improve the efficacy of their physical wellness, then a massive spike in cognitive health would follow shortly.

Due to the COVID-19 pandemic, a portion of the NHIS 2020 sample was replaced with a reinterviewed component of the 2019 sample adults [12]. There were three different weighting procedures used for this purpose: first, a weight for only the reinterviewed cases (a representation of the 2019 cases); second, a weight for the remaining 2020 cases representing a partial sample of the 2020 population; and third, a weight for the combined reinterviewed and remaining 2020 cases. The third dataset represents the 2020 population for analysis of 2020 sample adults and was used in this study. One limitation of the third procedure is that the biases are retained after the weighting adjustments. A consequence of this is an underrepresentation of adults living alone or in the lowest income category for the year 2020.

The study’s objective was to assess whether the ability to meet the recommended physical activity guidelines was associated with memory loss/difficulty concentrating in older adults living in the U.S.

## 2. Materials and Methods

This cross-sectional study used data from the 2020 National Health Interview Survey (NHIS). The authors utilized this survey codebook and data set to receive sampling and data on the desired test population. After selecting the dataset that had the highest frequencies to support the research question, the authors began developing our inclusion and exclusion criteria, which variables would be included and finally confirmed the study population/parameters. The NHIS is a nationally representative household survey of the U.S. (United States) civilian noninstitutionalized population [3]. Interviews were typically conducted in respondents’ homes, but follow-ups to complete interviews were conducted over the telephone. Information about the sample adult was self-reported unless physically or mentally unable to do so, and a knowledgeable proxy could answer for the sample adult. In 2020, there were 31,568 sample adult interviews, and the sample adult response rate for the 2020 sample, excluding reinterviewed sample adults, was 48.9%, while the response rate for the adult sample from the 2019–2020 longitudinal sample was 29.6%. The NHIS data are collected continuously from January to December each year.

### 2.1. Participants

The study population included adults aged 18+ living in the Northwest, Midwest, Western, and Southern United States who responded to the NHIS 2020 survey collected in 2019–2020. Participants who were physically or mentally unable to follow physical activity guidelines, those with a previous history of traumatic brain injury or cognitive impairment, those with a pre-existing physical condition other than heart disease, and those with missing information on any of the variables used in the study were excluded from the statistical analysis (*n =* 571). The final sample size was *n* = 30,119.

### 2.2. Variables

The main outcome variable was whether individuals had difficulty remembering/concentrating or both. This variable will be split into 5 categories: (1) no difficulty, (2) some difficulty + a lot of difficulty + cannot do at all, (3) difficulty remembering only, (4) difficulty concentrating only, and (5) difficulty with both. For the statistical analysis, categories 2–5 were merged into one single category (=at least some difficulties). The main independent variable was whether individuals met the aerobic and/or strength physical activity guidelines. This variable was split into 4 categories: (1) meets neither criterion, (2) meets strength only, (3) meets aerobic only, or (4) meets both criteria. The covariates that were tested in this analysis were race/ethnicity, body mass index (BMI), sex, age, history of heart disease, alcohol use, smoking status, household region, mental health status, and cancer status. Race was categorized into single/multiple race groups within Hispanic and was organized as follows: (1) Hispanic, (2) Non-Hispanic White only, (3) Non-Hispanic Black/African American only, (4) Non-Hispanic Asian only, and (5) Non-Hispanic American Indian Alaskan Native only + Non-Hispanic American Indian Alaskan Native and any other group. Other variables included body mass index, which was categorized as (1) underweight + healthy weight, (2) overweight, or (3) obese. Sex was categorized as (1) female or (2) male. Age had two variables: (1) 18+ and (2) 65+. A history of heart disease was considered; the authors only included individuals who answered no to (1) had coronary heart disease, (2) had angina, (3) had a stroke, or (4) had a heart attack. Coronary heart attack, angina, and stroke were combined into the “yes” category. Another included variable was alcohol use: (1) never/none, (2) per week, (3) per month, and (4) per year. Smoking status was also considered: (1) current daily smoker, (2) current some day smoker, (3) former smoker, or (4) never a smoker. Mental health status was also considered to indicate depression, or having depression was categorized as (1) yes or (2) no. Household region was important to note due to the environmental factors impacting one’s lifestyle choices: (1) Northwest, (2) Midwest, (3) South, or (4) West. Cancer was noted by respondents answering either (1) yes or (2) to having or previously having cancer.

### 2.3. Statistical Methods

Statistical Package for Social Science (SPSS) (version SPSSPERSONAL-MAC via Florida International University) was used to analyze the data. First, descriptive analysis was performed to attain a better understanding of the data. The authors then checked for distributions of variables and determined whether there was any missing information in our database. Next, the dichotomization of our data was performed due to the presence of only categorical variables, and a bi-variate analysis through STATA was also executed to identify any confounders, including chi-square tests for categorical variables, frequency distributions, and *t*-tests for continuous variables. This software was utilized when the authors relabeled all the variables and tested them for prevalence, frequency, and confidence intervals. Collinearity diagnostics were performed to assess the degree of correlation between variables. Finally, unadjusted and adjusted logistic regression analyses were conducted to calculate odds ratios (ORs) and corresponding 95% confidence intervals (CIs).

All data accessed were fully anonymized and without any of the 18 direct identifiers according to the Health Insurance Portability and Accountability Act. Ethical approval was waived by the Florida International University Health Sciences IRB since the analysis was considered non-human subjects research using de-identified data.

## 3. Results

Table 1 describes the distribution of U.S. population characteristics according to physical activity guidelines. All characteristics were differentially distributed according to physical activity guidelines. Whereas the highest percentage of ≥65-year-old participants was found among those who did not meet any of the PA guidelines criteria (25.9%), the prevalence of older adults was lowest among participants who met the target of the PA guidelines for both aerobic and strength (11.4%). Conversely, the highest percentage of <65-year-old participants met the PA guidelines (88.6%), while the prevalence was lowest in participants who met neither of the PA criteria (74.1%). There was a greater percentage of women (57%) among those who did not meet the PA guidelines than among those who met both criteria (44%). The corresponding percentage of women who met the strength criteria only or who met the aerobic criterion only were 50% and 51%, respectively. For men, a greater percentage of participants met both criteria (56.3%), and the prevalence was lowest for participants who met neither of the PA criteria (43.2%). The corresponding percentage of men who met the strength criteria only and those who met the aerobic criteria only were 50.3% and 49.2%, respectively. Those living below the poverty line (<1.00) had the highest prevalence of meeting neither of the PA criteria (13.1%), whereas the lowest prevalence met both PA criteria (5.37%). Conversely, participants 4–5 points above the poverty line had the highest prevalence (12.5%) of meeting both PA criteria and the lowest prevalence (10.3%) met neither PA criterion. Similarly, those with ≤5 points above the poverty line had the highest prevalence among those who met both PA criteria (43.2%), whereas the lowest prevalence was found among those who met neither PA criterion (22.1%). Those who answered “yes” to having or having had depression had the highest prevalence of meeting neither PA criterion (20.3%) and the lowest prevalence of meeting both PA criteria (11.7%). Those who did not or do not have depression had the highest prevalence of both PA criteria (88.3%) and the lowest in meeting neither PA criterion (79.7%). The corresponding strength criteria only and aerobic criteria only were 84.5% and 85.5%, respectively.

The association between difficulty following physical activity guidelines and cognitive difficulties was statistically significant (Table 2). Those who met aerobic only increased the odds of cognitive difficulty by 52% (OR 1.52; 95% CI: 1.34–1.74) compared with those who met both criteria. Those who met the strength criteria had 1.7-fold greater odds of having cognitive difficulties (OR 1.70; 95% CI: 1.42–2.02) than those who met both criteria. Those who met neither of these guidelines had almost threefold greater odds of having cognitive difficulties (OR 2.64; 95% CI: 2.36–2.96) than those who met both guidelines. Age was significantly more strongly associated with cognitive difficulty (OR 1.90; 95% CI: 1.73–2.10) in participants aged 65+ years old than in those aged <65 years old. Sex was not associated with cognitive difficulties (OR 0.97; 95% CI: 0.89–1.06). Participants living in the Midwest had significantly greater odds of having cognitive difficulties (OR 1.18; 95% CI: 1.01–1.37) than did those in the Northwest. The corresponding odds ratio of participants living in the West was 1.26 (OR 1.26; 95% CI: 1.09–1.44). Race/ethnicity was not associated with cognitive difficulties compared with NH White individuals, except for NH Asians, who had lower odds for cognitive difficulties (OR 0.70; 95% CI: 0.56–0.87). All levels of poverty variables were significantly associated with cognitive difficulties compared with those 5 points above the poverty line. Those living under the poverty line had increased odds of 2.22 (95% CI: 1.88–2.63). Those 1–2 points above the poverty line had 93% increased odds of having cognitive difficulties (OR 1.93; 95% CI: 1.68–2.23). A score of 2–4 points above the poverty line was also associated with cognitive difficulties (OR 1.57; 95% CI: 1.40–1.76). Participants 4–5 points above the poverty line had 1.32-fold increased odds (OR 1.32; 95% CI: 1.13–1.54). BMI was not associated with cognitive difficulties. Alcohol consumption was not significantly associated with cognitive difficulties compared to never drinking alcohol. Current drinkers had lower odds of having cognitive difficulties compared to former drinkers (OR 0.97; 95% CI: 0.82–1.13). Smoking was found to be significantly associated with cognitive difficulties. Compared with those who never smoked, current smokers had a 21% greater odds ratio (OR 1.21; 95% CI: 1.03–1.39). The corresponding odds ratio for former smokers was 1.29 (OR 1.29; 95% CI: 1.16–1.43). Heart disease (OR 1.37; 95% CI: 1.18–1.59), stroke (OR 2.38; 95% CI: 1.98–2.89), cancer (OR 1.24; 95% CI: 1.10–1.40), and depression (OR 4.69; 95% CI: 4.25–5.17) were also found to be associated with cognitive difficulties.

## 4. Discussion

Our data revealed that those who could not meet physical activity guidelines had a greater prevalence of cognitive difficulties. Moreover, compared with participants who met both the aerobic and the strength exercise guidelines, those who met none of the targets or met only one component of the guidelines had greater odds of difficulty remembering/concentrating. In addition, poverty level, depression, and heart disease were associated with increased odds of cognitive difficulties.

Our findings are in line with those of previous studies [1,2,7,8,9,10,11,12]. A study was conducted on 25 middle-aged adults from the Wisconsin Registry for Alzheimer’s Disease (AD) Prevention. This study compared biomarkers from participants who performed enhanced physical activity with plasma biomarkers prior to starting the exercise regimen. They found that exercise improved cognition and overall learning/memory abilities based on the respiratory exchange ratio, the perceived exertion rating, and plasma biomarker measurements [9]. The Wisconsin Registry study revealed results that physical activity improved cognition, learning, and overall memory [9]. Similarly, a population-based cohort study that measured the association between a healthy lifestyle and memory decline in older adults who had normal cognition revealed that a healthy lifestyle was associated with slower memory decline based on linear mixed effects models with the standard z-score of the Mini-Mental State Examination. After an additional 10-year follow-up, their original findings were consistent, showing that participants in the favorable group had slower memory decline than those in the unfavorable group. They also found that individuals with an average lifestyle exhibited a slower memory decline than those with an unfavorable lifestyle [11]. This population-based cohort study had similar findings to our study in that a healthy lifestyle (composed of exercise and dieting) resulted in slower memory decline and improved cognition in older adults. The 10-year follow-up confirmed these results. A cross-sectional study of the associations between physical fitness and cognitive function in community-dwelling older adults sampled 107 older people. They reported that there was an association between physical fitness and cognitive function, specifically stating that the grip strength and the 6 min walk were positively related to cognitive function, with an *r* = 0.42 and 0.35, *p* < 0.05, while the 5-repetition sit-to-stand test was negatively associated with cognitive function with an *r* = −0.43, *p* < 0.01 [12]. Conversely, our study was population-based (U.S. adults aged 18+ and older adults aged 65+) and compared the percentage of self-reported cognitive difficulties between participants who met various physical activity criteria. Their findings showed that physical activity had a positive impact on the cognitive function of community-dwelling older adults. All these studies revealed a positive correlation between physical activity and cognitive function.

Several mechanisms have been proposed for how physical activity may protect against memory loss [13,14,15,16]. Neuroimaging data have shown that aerobic fitness prevents the loss of aging-related brain tissue and enhances the structural integrity of memory-related brain areas [14]. This finding highlights the positive impact of physical activity on cognitive function. Furthermore, a similar study also revealed that physical exercise is a gene modulator that induces structural changes in the brain, promoting cognitive benefits [15]. Similar to our study, Gomez-Pinilla et al. measured a form of physical activity in aerobic exercise and found that it induces positive brain function and energy metabolism, further supporting the idea that activity leads are related to improved cognitive function. Gomez-Pinilla et al. focused on how aerobic exercise triggers the activation of other regions of the brain, specifically, how exercise directly correlates with molecular events and energy metabolism, and Mandelosi et al. focused on physical activity as a benefit to biological and psychological well-being. It has also been established that brain-derived neurotrophic factor (BDNF) is positively associated with cognitive performance [7,9,16]. BDNF is a molecule in the brain that responds to plasticity changes that impact learning and memory. A study reported that molecules released after exercise can induce the expression of necessary promoters of the BDNF gene [17]. Furthermore, Castaño et al. reported that resistance training combined with cognitive training increased BDNF and had positive impacts on cognitive ability in general [7]. This finding further illustrates that exercise has a positive impact on the brain’s production of certain proteins and neurochemicals, which results in improved cognition. Aerobic exercise on treadmills combined with BDNF cognitive exercises has also been shown to increase neurogenesis [6]. They found that adults with cognitive impairments have lower Cathepsin B (CTSB) levels in the brain, and aerobic exercise training elevated these levels and improved memory function [6]. Similarly, KLOTHO, a protein that enhances cognition and synaptic function, was found to be upregulated by exercise and is thought to support structures related to memory and learning [9]. Moreover, resistance exercise (RE) can cause beneficial changes in blood flow, neurotransmission, and endocrine metabolism and promote cerebrovascular regeneration, all of which contribute to a delay in the cognitive degradation commonly observed with age [18]. Cheng described how RE can stimulate different neurochemicals, impacting the neurobiological processes and cognition through environmental changes. One example of this is insulin growth factor (IGF) and growth hormone (GH), both of which decrease with age. Sufficient IGF-1 levels in the blood are needed for precursor BDNF proteins to form, leading to BDNF production. Thus, IGF-1 is crucial for cognition [19]. After 52 weeks of RE, the concentration of IGF-1 in elderly individuals increased, and cognitive function improved, as indicated by recall and reaction times [19]. Chronic neuroinflammation is associated with the cognitive decline seen in Alzheimer’s Disease (AD^)^ [1]. Through RE intervention, peripheral and encephalitis cytokines are downregulated, inhibiting the expression of pro-inflammatory factors and regulating microglial activation. This prevents the oxidation and inflammatory mechanisms observed in AD [18,19].

Those who had depression were five times more likely to have cognitive deficits than those who did not. Codella et al. revealed that regular practice of physical activity reduces one-quarter of all depressive episodes and that physical activity and depressive episodes are inversely related [20]. Pseudodementia is a term used to describe cognitive deficits that often occur during depressive episodes and can even cause irreversible changes [21]. Even after remission of depressive episodes, families of individuals note cognitive decline weeks after an episode, including memory loss and difficulty paying attention [21]. Cognitive impairment is extremely prominent in individuals with major depressive disorder [22]. They found that these paradigms help assess cognitive ability, specifically the diagnosis and impact of depression, in their sample population. Furthermore, the understanding of the neurobiology of depression diagnoses must be expanded, and these improved parameters will greatly help in understanding the clear cognitive deficits reflected by MDD. Our data revealed that living at or below the poverty line increased one’s risk of cognitive difficulties twofold. Poverty has been reported to be related to socioeconomic status and the surrounding environment, which impacts neurodevelopment [23]. Stress can negatively impact brain health, and low income is positively associated with increased stress levels [23]. Previous studies also revealed a correlation between decreased surface area of gray matter in the frontal and temporal regions in children from families living below the poverty line [23]. This is thought to be due to increased stress associated with poverty, a chaotic household life, nutritional deficits, and neglectful caregivers [23]. This further contributes to cognitive difficulties in adulthood. A study of Chinese adults revealed that those who lived in poverty for a longer duration had lower CMMSE scores. This further reinforces the point that income level and poverty directly negatively impact cognitive ability. Subsequently, it has been reported that poverty negatively impacts cognitive development and impairment [24]. Children with greater levels of neighborhood poverty scored notably lower across all cognitive domains [24]. Furthermore, the volume scores of the dorsolateral prefrontal cortex, dorsomedial prefrontal cortex, and superior frontal gyrus were also lower due to greater neighborhood poverty, further bolstering the claims outlined in our study regarding poverty’s negative grasp on brain health and cognitive ability [24].

Naturally, our study has several limitations. Due to the cross-sectional research design, there is little evidence of causal associations. There is no evidence that cognitive difficulties occurred after a lack of physical activity or prior to physical activity. Our study design disables our ability to measure the impact that time has on the development of cognitive difficulties following the inability to meet the physical activity guidelines. Moreover, the questionnaires used in the study involved self-reported responses pertaining to hours spent being physically active, introducing some reporting bias. Additionally, residual confounding factors may bias the association between meeting the physical activity guidelines and cognitive difficulties, as we did not have information on a family history of cognitive decline or dementia.

## 5. Conclusions

In conclusion, physical activity is a modifiable lifestyle factor that can decrease one’s risk of cognitive difficulties. This study highlights the significance of the association between meeting physical activity guidelines and cognitive performance. The results indicate a 3-fold greater odds of developing cognitive difficulties in those who did not meet any physical activity guidelines compared to those who met the recommended physical activity guidelines. Physicians and public health professionals should prioritize reinforcement and encouragement to meet these guidelines. All ranges of poverty levels were significantly associated with increased odds of cognitive difficulties. Considering the association between poverty levels and the inability to meet the physical activity guidelines, improved access to healthcare providers and physical activity spaces should be targeted. National efforts to emphasize the significance of environmental factors and how facilities are built in an area may lack the necessary commodities to promote healthier lifestyle practices. Healthcare providers’ aim to elaborate upon the weight of physical activity and its impact on cognition should also be examined. Local leaders should attempt to provide equal access to physical activity facilities for lower socioeconomic status (SES) populations. Moreover, local physicians should bolster the recommendations made for physical health and further reinforce the significance of different forms of physical activity on brain health. Finally, future studies should look at physical activity as an intervention method for those with a genetic predisposition to dementia. Furthermore, studies should aim to develop a deeper mechanistic understanding of the positive relationship between physical activity and cognition.

## Figures and Tables

**Table 1 ijerph-21-01193-t001:** Baseline characteristics of U.S. adults by physical activity guidelines (NHIS 2020).

**Characteristics**	**Meets Neither Criteria**		**Meets Strength Only**		**Meets Aerobic Only**		**Meets Both Criteria**	
	*n*	%	*n*	%	*n*	%	*n*	%
**Age (years)**								
Less than 65	8570	74.1	1387	79.4	4836	76.9	5820	88.6
65 or older	5054	25.9	653	20.6	2429	23.1	1370	11.4
**Sex**								
Male	5573	43.2	954	50.3	3421	49.2	3897	56.3
Female	8051	56.8	1085	49.7	3844	50.8	3293	43.7
**Household Region**								
Northwest	2385	17.2	366	16.9	1295	18.1	1275	17.6
Midwest	3228	21.2	455	20.1	1692	21.8	1554	20.3
South	5008	40.4	702	39.2	2358	35.4	2258	34.2
West	3013	21.2	517	23.8	1920	24.7	2103	27.9
**Race/Ethnicity**								
Hispanic	1787	18.3	248	17.8	779	14.7	829	15.4
NH White	9279	60.4	1382	59.1	5347	67.2	5138	65.5
NH Black	1569	12.6	230	13.7	574	9.34	620	10.6
NH Asian	647	5.62	122	6.58	415	6.53	425	5.77
NH Other	342	3.03	58	2.75	150	2.22	178	2.67
**Poverty (poverty line is 1.00)**								
<1.00	1632	13.1	161	8.71	517	8.14	328	5.37
1.00 to 1.99	2844	21.9	328	18.3	1048	16.3	685	11.7
2.00 to 3.99	4391	32.7	582	29.7	2087	29.7	1785	27.3
4.00 to 4.99	1458	10.3	262	12	895	11.8	916	12.5
5.00 or higher	3299	22.1	707	31.2	2718	34	3476	43.2
**BMI**								
Underweight	223	1.76	26	1.41	96	1.43	84	1.09
Healthy	3404	25.8	632	31.7	2398	33.6	2903	39.7
Overweight	4268	31.8	707	34.3	2634	34.8	2674	37.3
Obese	5341	40.6	644	32.6	2006	30.2	1459	21.9
**Alcohol**								
Never	1820	16	184	12.7	704	12.5	515	9.56
Former	3289	21.7	403	18.1	1251	16	794	10.5
Current	8392	62.2	1443	69.2	5265	71.5	5836	80
**Smoke**								
Current	2005	15.6	211	10.5	785	12	473	7.12
Former	3672	23.5	537	23.1	1994	23.8	1675	19.8
Never	7931	60.9	1288	66.4	4476	64.2	5033	73.1
**Heart Disease**								
Yes	1401	8.13	154	5.69	493	5.49	269	2.66
No	12,178	91.9	1881	94.3	6761	94.5	6916	97.3
**Stroke**								
Yes	617	3.62	87	3.32	176	1.94	99	0.986
No	12,992	96.4	1952	96.7	7087	98.1	7088	99
**Diabetes**								
Yes	2004	12.7	202	8.19	628	7.78	307	3.52
No	11,608	87.3	1836	91.8	6632	92.2	6879	96.5
**Depression**								
Yes	2876	20.3	371	15.5	1131	14.5	882	11.7
No	10,726	79.7	1667	84.5	6126	85.5	6298	88.3
**Cancer**								
Yes	1944	10.8	297	9.84	992	10.1	684	6.52
No	11,662	89.2	1743	90.2	6268	89.9	6504	93.5

**Table 2 ijerph-21-01193-t002:** Associations between the characteristics of the study participants and cognitive difficulties in U.S. adults 18+.

**Characteristics**	**Unadjusted**	**Adjusted**
	**OR ^1^ (95% CI ^2^)**	**OR (95% CI)**
**Physical Activity**		
Meets neither criteria	** 2.64 (2.36–2.96)	1.69 (1.48–1.92)
Meets strength only	** 1.70 (1.42–2.02)	1.29 (1.05–1.57)
Meets aerobic only	** 1.52 (1.34–1.74)	1.21 (1.05–1.40)
Meets both criteria	Reference	Reference
**Age (Years)**		
Less than 65	Reference	Reference
65 or older	** 2.23 (2.06–2.40)	1.90 (1.73–2.10)
**Sex**		
Male	Reference	Reference
Female	1.25 (1.16–1.35)	0.97 (0.89–1.06)
**Household Region**		
Northwest	Reference	Reference
Midwest	* 1.30 (1.13–1.50)	1.18 (1.01–1.37)
South	1.19 (1.05–1.34)	1.04 (0.91–1.18)
West	1.14 (1.0–1.31)	1.26 (1.09–1.44)
**Race/Ethnicity**		
Hispanic	0.76 (0.69–0.89)	0.87 (0.75–1.01)
NH White	Reference	Reference
NH Black	0.96 (0.84–1.10)	1.04 (0.90–1.21)
NH Asian	* 0.51 (0.41–0.63)	0.70 (0.56–0.87)
NH Other	1.25 (0.94–1.66)	1.17 (0.84–1.62)
**Poverty (poverty line is 1.00)**		
<1.00	3.17 (2.75–3.65)	2.22 (1.88–2.63)
1.00 to 1.99	2.62 (2.32–2.95)	1.93 (1.68–2.23)
2.00 to 3.99	1.88 (1.68–2.10)	1.57 (1.40–1.76)
4.00 to 4.99	1.48 (1.28–1.72)	1.32 (1.13–1.54)
5.00 or higher	Reference	Reference
**BMI ^3^**		
Underweight	1.77 (1.33–2.34)	1.34 (0.96–1.88)
Healthy	Reference	Reference
Overweight	1.02 (0.93–1.13)	0.92 (0.83–1.02)
Obese	1.35 (1.22–1.49)	0.97 (0.86–1.09)
**Alcohol**		
Never	Reference	Reference
Former	1.69 (1.47–1.97)	1.18 (1.00–1.39)
Current	0.91 (0.79–1.04)	0.97 (0.82–1.13)
**Smoke**		
Current	* 1.85 (1.66–2.06)	1.21 (1.05–1.39)
Former	1.71 (1.56–1.88)	1.29 (1.16–1.43)
Never	Reference	Reference
**Heart Disease**		
Yes	* 2.74 (2.42–3.10)	1.37 (1.18–1.59)
No	Reference	Reference
**Stroke**		
Yes	* 4.57 (3.91–5.33)	2.38 (1.95–2.89)
No	Reference	Reference
**Diabetes**		
Yes	2.00 (1.80–2.22)	1.11 (0.99–1.25)
No	Reference	Reference
**Depression**		
Yes	* 5.12 (4.69–5.58)	4.69 (4.25–5.17)
No	Reference	Reference
**Cancer**		
Yes	1.87 (1.69–2.06)	1.24 (1.10–1.40)
No	Reference	Reference

^1^ odds ratio; ^2^ confidence interval; ^3^ body mass index. * statistically significant ** variables of focus.

## Data Availability

Data are available upon reasonable request.

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
