# Peer review of "The Impact of Physical Activity on Memory Loss and Concentration in Adults Aged 18 or Older in the U.S. in 2020"

_ijerph, 2024, doi:10.3390/ijerph21091193_

Round 1

Reviewer 1 Report

Comments and Suggestions for Authors

The chosen title is formulated as a question and is very long, failing to clearly capture the main purpose of the article. Therefore, I recommend choosing a shorter title that tells the purpose of this research at a glance and contains at least three of the keywords.

Given that dementia is an umbrella term, used to describe several neurological conditions, similar through a similar clinical picture, which includes manifestations such as memory impairment, cognitive disorders and alteration of social integration skills, please reconsider and change  "dementia symptoms/cognitive difficulties" (line 15-16). 

The introduction tries to present the current state of knowledge of the subject of this study, but we consider that only 9 bibliographic references are not enough for this. Therefore please go back to the introduction and add other existing bibliographic sources that will ensure a better representation of current knowledge.

The conclusions are write without making direct reference to the results obtained , so, please supplemented each statement in the conclusions with direct references to the results obtained from the research.

The number of bibliographic resources is too small to support this study. Please completing the bibliographic list with at least 10 titles.

Author Response

Comment #1: Title is very long, formulated as a question, and failing to capture main purpose of article. Suggests a shorter title containing at least three of the keywords.  

Response #1: The title has been adjusted to better capture the main idea of the article and include keywords. We also followed reviewer #4’s suggestion. 

Previous title:  

Is having difficulty meeting the recommended aerobic and strength physical activity guidelines associated with memory loss/difficulty concentrating in adults aged 18 or older in the US in 2020? 

New recommended title: 

“The impact of physical activity on memory loss of concentration in adults aged 18 or older in the US in 2020” 

Comment #2: please reconsider and change "dementia symptoms/cognitive difficulties" (line 15-16) as it is an umbrella term and too broad.  

Response #2: We have now changed lines 15-16 to be more specific.  

Line 15-16: “The main outcome was whether individuals experienced dementia symptoms such as memory loss and difficulty concentrating”.  

Comment #3: ... only 9 bibliographic references are not enough, please add other existing sources to ensure a better representation of current knowledge 

Response #3: We have now added more existing sources to represent current knowledge. See lines 52-56 

Comment #4: Conclusions should make direct references to the results obtained, supplement each statement in conclusions with direct references to the results obtained from the research 

Response #4: Please refer to lines 291-292, 294-295, and for suggested direct references to results. 

Comment #5: Number of bibliographic resources is too small to support this study, must have at least 10 titles 

Response #5: We have increased our sources to 10 titles, see references section.  

Reviewer 2 Report

Comments and Suggestions for Authors

Dear Authors

Your work is important, and I appreciate your effort and dedication to carrying out this study. Below, I leave comments on some aspects that I hope will help improve your manuscript:

Abstract: I suggest providing more information about the sample studied, such as the sample size, percentage by sex, and average or age range. The country of the survey should be indicated. The study's objective and the background leading to that objective should also be indicated. Regardless of whether the data are secondary, information should be provided about the methodology used, including the variables and measurement instruments. The format should be structured.

Introduction: Your work's absence of a precise knowledge gap underscores the necessity of enriching the state of the art with studies that delimit what is known in this line of research. This is where your contribution becomes crucial. On the other hand, it would also be desirable to incorporate hypotheses following the stated study objective. I suggest incorporating these hypotheses from the state of the art. Having hypotheses will help guide your discussion.

Methods: The information provided in the methodology must be reorganized through subheadings as suggested by the journal format. The inclusion and exclusion criteria of the sample used must be specified, as well as the quantity eliminated by the criterion. In addition, the ethics committee body that approved the study must be indicated, regardless of whether the study is based on secondary data.

Statistical analysis section: The groups used in Table 1 and the criteria for separation should be mentioned and described here. Regarding the comparison analysis performed, it should be noted that although a sample size that is too large, as in this case, increases statistical power and decreases type II error, there is also a greater risk of identifying statistically significant differences that may not be relevant in practice, potentially increasing type I error. Therefore, it is not recommended to perform this type of analysis. I suggest omitting this analysis and reporting Table 1 without differences between groups. On the other hand, in the logistic regression analysis, the variables included in the adjusted and unadjusted model should be specified.

Table 1: The descriptive paragraph in Table 1 is excessively long. The main findings should be reported here in an interpretive manner, highlighting differences between groups or any type of pattern that characterizes them. The table must indicate the sample size per group and what percentage of the total sample it corresponds to.

Table 2: The details of the model must be specified. It is understood that the relationship between physical activity and cognitive difficulty is to be examined, with compliance with physical activity recommendations being the predictor variable of cognitive ability. Therefore, different models should be created for each physical activity compliance group, and then different adjustments should be made to each model depending on the variables added. However, this table makes interpretation difficult since the models are not specified.

Discussion: The discussion should be reviewed in light of the adjustments made to the analyses. This is a crucial part of your work, as it is not enough to mention recent evidence in relation to the topic studied. It is necessary to incorporate a contrast of the findings with the reported evidence, generating an interpretation of the results of your study.

Author Response

Comment #1:  

Abstract: Provide more information about sample studied, such as sample size, percentage by sex, and average or age range. Indicate country of survey. The study’s objective and background leading to said objective should also be indicated. Information should be provided about methodology used, including variables and measurement instruments. Format should be structured.  

Response #1: Please refer to the highlighted sections in the abstract to see the adjustments made.  

Introduction: Absence of precise knowledge gaps. Please address to emphasize why contribution to this field is critical. Incorporate hypothesis following the stated study objective.  

Response #2: please see highlighted sections in introduction, in which we emphasize the importance of further contributing to this area of study.  

Methods: Information in methodology should be reorganized with subheadings. The inclusion and exclusion criteria of the sample used must be specified plus the quantity eliminated by the criterion. Indicate the ethics committee body that approved the study, even if secondary data 

Response #3: please see new highlighted section subheadings in the document 

Statistical analysis section: The groups used in Table 1 and the criteria for separation should be mentioned and described here. Please note that while a large sample size increases statistical power and decreases type 2 error, there is also greater risk of identifying statistically significant differences that may not be relevant in practice, potentially increasing type 1 error. Therefore, not recommended to perform comparison analysis as such. Omit and report table 1 without differences between groups. For the logistic regression analysis, variables included in adjusted and unadjusted model should be specified.  

Response #4: The tables are separated in such a way with the intent to inform audiences about the sample characteristics and data analysis. The comparison analysis was done to outline the impact of physical activity on cognitive ability across different populations. Many other studies failed to incorporate the statistical weight of poulation variety, the comparison aided in outlining specificityThe adjusted and unadjusted data regarding the variables are all within the tables prior to the analytical breakdown for the logistic regression.  

Table 1: descriptive paragraph is excessively long. Main findings should be reported here in interpretive manner, highlight differences between groups or any type of pattern that characterizes them. Table must indicate sample size per group and percentage of total sample it corresponds to.  

Response #5: The main findings discuss and explain the differences between groups and the trends found in the data. The percentages are in the tables to the right.  

Table 2: Specify details of model. Different models for each physical activity compliance group should be created and then different adjustments should be made to each model depending on the variables added. Table makes interpretation difficult since models are not specified.  

Response #6: Given the resources available at the time of data collection and interpretation, a new model for each group is not feasible. However, for future studies this will be taken into consideration.  

Discussion: Should be reviewed in light of adjustments made to analyses. Incorporate a contrast of findings with reported evidence, generating an interpretation of the results of study.  

Response #7: We included a comparative breakdown within the discussion portion of our research article. This section included comparisons to other studies sharing both similar and totally different results. There were only minor adjustments made to the article and thus warranted minimal change in our discussion. More guidance is needed on executing the comments. 

Reviewer 3 Report

Comments and Suggestions for Authors

Dear editor of the journal. Thank you for choosing me as a reviewer.

The title can be written more interesting and attractive.

The introduction is very sparse. The statement of the problem is ambiguous and the necessity of conducting the research is unclear. For example, Covid is mentioned in the introduction, but there is no connection between the subject.

In the working method, how was the method of sampling and receiving information.

In the results section, it is better to use the figure for comparison

In the discussion, due to the many variables, it was expected to have a more comprehensive discussion and to mention the reasons for effectiveness in full.

Author Response

Title: Can be more interesting and attractive 

Response #1: We have adjusted the title, please see the highlighted new title in the attached manuscript.  

Introduction: very sparse. Statement of problem is ambiguous and necessity of conducting research is unclear. For example, Covid is mentioned in the introduction, but there is no connection between the subject. 

Response#2: See highlighted sections for corrections. As for the COVID portion, it was stated to outline how the pandemic altered our sample and data pool. It was not tied to dementia. 

Methods: how was the method of sampling and receiving information 

Response#3: See highlighted sections for corrections. As for the COVID portion, it was stated to outline how the pandemic altered our sample and data pool. It was not tied to dementia. 

Results: it is better to use figure for comparison 

Response#4: The tables provided highlight comparisons, for future posters and/ or abstracts we agree that figures for comparison would be good to consider.  

Discussion: due to the many variables, it was expected to have a more comprehensive discussion and to mention the reasons for effectiveness in full. 

Response #5: Please refer to discussion section for adjustments.  

Reviewer 4 Report

Comments and Suggestions for Authors

Thank you for submitting send manuscript “Is having difficulty meeting the recommended aerobic and strength physical activity guidelines associated with memory loss/difficulty concentrating in adults aged 18 or older in the US in 2020?”. Presenting the research results of the submitted scientific article required many sacrifices and logistics and organization of scientific research. After reviewing mentioned above scientific paper I would like to require many advice and changes.

Below I am sending You outline possible points for revision in the chronological order of the manuscript.

Title. I propose that the topic of the research paper should be changed to “The impact of physical activity on memory loss of concentration in adults aged 18 or older in the US in 2020”

Introduction. Lack of citation in the line 35. In the introduction section, the authors often use abbreviations which the authors do not explain. Please explain what the following abbreviations mean: BDF, WHO (for readers), NHIS. Lack of citation in the line: 60 and 62. the introduction should be more elaborate. The authors should show how important physical activity is especially for older people. While reading the introduction, it feels as if the authors did not make use of the many scientific publications that describe how important physical activity is for people of all ages. In addition, the authors should show by quoting other publications what affects dementia, how the disease is being tackled these days.

Material and methods: The material and methods section is too long. The authors should focus on the most important information for this section. In addition, I propose a subdivision into subsections 2.1 Participants, 2.2 Statistical methods. Line 97 - I propose to correct the sentence in English. Furthermore, the authors do not only use the following phrases in this section: we tested, we included, we checked - I propose to use impersonal phrases not only in this section for example It was tested, The authors included and so on....

Results. The findings are clear. One comment only for table number 2 Meets Neither Criteria - should be Meets neither criteria.

Discussion. The most scientifically weakest section is the discussion section. In the discussion, the authors use very few citations or publications with which to discuss. Very often authors use one publication and describe it in its entirety. The discussion should be based on many scientific publications because then the article can be evaluated positively. The discussion needs to be restructured by citing definitely more scientific publications on the topic of the scientific article.

References. In the literature, the authors should correct the following citations (missing pages): 3,4,5,7,9,10,15,16,21. In addition, a scientific article in which the authors have used only 21 citations/publications must be corrected. The problem of dementia and physical inactivity is global and the authors should definitely work on a literature review on this topic. More scientific publications will help the authors to improve the introduction and discussion in the submitted scientific article.

Author Response

Title: I propose that the topic of the research paper should be changed to “The impact of physical activity on memory loss of concentration in adults aged 18 or older in the US in 2020” 

Response #1: We followed reviewer 4’s suggestion. Please see title change 

Introduction: lack of citation in the line 35. Please explain all abbreviations such as the following: BDF, WHO, NHIS. Lack of citation line 60 and 62. Introduction should be more elaborate. The authors should show how important physical activity is especially for older people. It feels as if the authors did not make use of the many scientific publications that describe how important physical activity is for people of all ages. Quote other publications what affects dementia, how the disease is being tackled these days.  

Response#2: Please see highlighted portions, added a new reference regarding physical activity on older adults, particularly for older adults. Also corrected abbreviation errors and added a paragraph on the significance of physical activity on older adults. 

Materials and Methods: This section is too long. Focus on the most important information. I propose a subdivision into subsections 2.1 Participants, 2.2 Statistical methods. Line 97 - I propose to correct the sentence in English. Furthermore, the authors do not only use the following phrases in this section: we tested, we included, we checked - I propose to use impersonal phrases not only in this section for example It was tested, The authors included and so on.... 

Response #3: Corrected the impersonal phrases. Line 97 was corrected as well. Split the methods portion into 3 more sections 2.1-2.3. Shortened the methods section significantly. 

Results: Results are clear. For table number 2 Meets Neither Criteria- should be Meets neither criteria 

Response #4: Corrected table 2 typing error.  

Discussion: Scientifically weak. Use more citations/ publications to discuss. Needs to be restructured by citing definitely more scientific publications on the topic of the scientific article.  

References: Correct the following citations (missing pages): 3,4,5,7,9,10,15,16,21. More scientific publications to help both discussion and introduction.  

Response #5: Please refer to references to see adjustments.  

Round 2

Reviewer 2 Report

Comments and Suggestions for Authors

Dear authors,

You have substantially improved your manuscript. You need to specify the variables and their respective categories used to create the adjusted logistic regression model. This should be detailed in the statistical analysis section. On the other hand, performing t-tests or chi-square tests on large sample sizes can be misleading since small differences may be significant due to the large sample size. Therefore, this analysis should be discarded from Table 1.

The information provided in lines 146-148 corresponds to the section on participants. In addition, they must include which organization or ethics committee approved the study (National Health Survey). It would also be desirable to indicate whether they have followed ethical guidelines, such as the Declaration of Helsinki.

Author Response

Comment #1: You need to specify the variables, and their respective categories used to create the adjusted logistic regression model. This should be detailed in the statistical analysis section. On the other hand, performing t-tests or chi-square tests on large sample sizes can be misleading since small differences may be significant due to the large sample size. Therefore, this analysis should be discarded from Table 1. 

Response #1: All the variables and their respective categories are now described in detail in section 2.2. in agreement with your suggestion, we have now removed the p-values from Table 1. 

Comment #2: The information provided in lines 146-148 corresponds to the section on participants. In addition, they must include which organization or ethics committee approved the study (National Health Survey). It would also be desirable to indicate whether they have followed ethical guidelines, such as the Declaration of Helsinki. 

Response #2: We have revised the ethical statement accordingly on page 10 of the manuscript (lines 319-321). The ethical statement (institutional review statement) reads now as follows: 

The study was conducted in accordance with the Declaration of Helsinki. This study received an exemption for review from the Institutional Review Board at Florida International University as NHIS data are publicly available and deidentified [24]. 

The following reference was added: 

 [24] IRB Exemption. [Accessed on September 21st, 2023] Available from: https://www.hhs.gov/ohrp/regulations-and-policy/decision-charts-2018/index.html. 

Reviewer 3 Report

Comments and Suggestions for Authors

Article edited and its can be publish

Author Response

Comment #1: Article edited and it can be published 

Response #1: Thank you :)

Reviewer 4 Report

Comments and Suggestions for Authors

Thank you very much for sending a revised version of the scientific article. The authors of the scientific article have greatly enriched the scientific value of the manuscript. Most of the suggestions or corrections have been made. However, the manuscript should still be improved with the following things: 

The authors should still improve the discussions. In the first version of the scientific article there were 21 citations, in the second version the authors added only 2 publications. It is reasonable to add definitely more publications. Furthermore, while reading the discussions, the authors very often rely on one publication: line 245-256 (publication 17), (line 262-268 or 268-279). The authors should increase the number of citations in the discussion related to the topic of the scientific article.

Author Response

Comment #1: The authors should still improve the discussions. It is reasonable to add definitely more publications. Furthermore, while reading the discussions, the authors very often rely on one publication: line 245-256 (publication 17), (line 262-268 or 268-279). The authors should increase the number of citations in the discussion related to the topic of the scientific article. 

Response #1: Thank you for your suggestions.  We have further improved the discussion section by adding more references per discussion points. All changes have been marked in the text with yellow highlights.